# Backscattering in topological edge states despite time-reversal symmetry

Jonas Erhardt[1,2], Mattia Iannetti [3], Fernando Dominguez[2,4], Ewelina M. Hankiewicz[2,4], Björn Trauzettel [2,4], Gianni Profeta[3,5], Domenico Di Sante [6], Giorgio Sangiovanni [2,4], Simon Moser [1,2] & Ralph Claessen [1,2] ✉

Spin-momentum-locked edge states of quantum spin Hall insulators provide a compelling platform for spintronic applications, owing to their intrinsic protection against backscattering from non-magnetic disorder. This protection emerges from time-reversal symmetry, which pairs Kramers partners of helical edge modes with opposite spin and momentum, thereby strictly forbidding elastic single-particle backscattering within the pair. Yet, contrary to the idealized notion of linear edge bands, the non-monotonic dispersions of realistic materials can host multiple Kramers pairs, reintroducing backscattering channels between them without violating time-reversal symmetry. Here, we investigate inter-Kramers pair backscattering in the non-linear edge bands of the quantum spin Hall insulator indenene, highlighting a critical aspect of edge state stability. Using quasiparticle interference in scanning tunneling spectroscopy – a direct probe of backscattering – we observe intra-band coupling between different Kramers pairs, while energy regions with only a single Kramers pair remain strictly protected. Supported by theoretical analysis, our findings provide an unprecedented experimental demonstration of edge state backscattering fully consistent with their underlying topological protection. This insight has profound implications for numerous quantum spin Hall insulator candidates, emphasizing that the mere presence of gaptraversing edge modes does not inherently guarantee their protection against backscattering.

Dissipationless ballistic transport in spin-polarized metallic edge modes of quantum spin Hall insulators (QSHIs) forms the foundation of their technological potential[1–6] and sparked an intense search for new topological materials[7,8]. In a QSHI, time-reversal symmetry (TRS) and the resulting spin-momentum locking protect Kramers partners−time-reversal symmetric edge modes forming a Kramers pair−from elastic single-particle backscattering. This protection occurs because the momentum transfer $q$ required for backscattering would necessitate a TRS-violating spin flip, which is therefore not allowed as indicated in Fig. 1a$_1$[9]. As a result, both modes are perfectly transmitted at nonmagnetic defects and propagate freely along the edge when they are equally populated (Fig. 1a$_2$). In the non-equilibrium situation of electronic transport this would lead to a two-terminal edge conductance of $2e^2/h$[10], where the factor of 2 accounts for the contributions from both edges.

[1]Physikalisches Institut, Universität Würzburg, D-97074 Würzburg, Germany. [2]Würzburg-Dresden Cluster of Excellence ct.qmat, Universität Würzburg, D-97074 Würzburg, Germany. [3]Department of Physical and Chemical Sciences, University of L'Aquila, Via Vetoio, 67100 L'Aquila, Italy. [4]Institut für Theoretische Physik und Astrophysik, Universität Würzburg, D-97074 Würzburg, Germany. [5]SPIN-CNR, University of L'Aquila, Via Vetoio 10, 67100 L'Aquila, Italy. [6]Department of Physics and Astronomy, University of Bologna, 40127 Bologna, Italy. ✉e-mail: claessen@physik.uni-wuerzburg.de

**Fig. 1 | Backscattering between helical edge states.** $a_1$ Schematic linear edge dispersion of a QSHI with one single Kramers pair within the topological bulk gap[10]. Blue and red colors mark channels of opposite spin polarizations. $a_2$ Corresponding real-space cartoon of a right-moving spin up current (blue: positive group velocity $dE/dk$) and a left-moving ($dE/dk < 0$) spin down current (red) allocated at a single edge of a QSHI. Scattering between these channels is prohibited by TRS. $b_1$ Schematic non-linear but $\mathcal{S}$-shaped edge dispersion of a QSHI with energy regions of one and three Kramers pairs within the topological bulk gap, the latter indicated by purple shading. $b_2$ Real-space cartoon of the three Kramers pairs scenario, with disparate numbers of left and right movers in the two spin channels. TRS-conserving backscattering between two Kramers pairs at non-magnetic defects produces standing waves with wavelength $2\pi/q$.

Historically, this scenario was discussed mostly for linear dispersing edge bands of ideal honeycomb materials[10–12] and quantum wells[13–15] that produce only a single Kramers pair (Fig. 1$a_1$). However, this picture is often too simplistic given that in real QSHI materials a multi-orbital character[16], buckling[17], or details in the edge potential[18,19] render the band dispersion of edge modes more complicated, typically highly non-linear[1,5,20–22]. Without repercussions on the topological $\mathcal{Z}_2$ invariant and the bulk-boundary correspondence, such edge-specific modifications to the Hamiltonian can, in principle, produce an $\mathcal{S}$-shaped edge dispersion, resulting in a finite energy interval with an odd number (here: $n = 3$) of degenerate Karmers pairs within the bulk gap (Fig. 1$b_1$). In contrast to the pure linear band scenario of Fig. 1a, the edge conductance in such systems should thus be either $2e^2/h$ or $6e^2/h$, depending on whether the precise position of the Fermi level activates $n = 1$ or $n = 3$ Kramers pairs (Fig. 1$b_1$)[23,24].

However, while for the $n = 1$ band section, TRS continues to strictly prohibit elastic scattering, the $n = 3$ section contains right- and left-moving modes of identical spin character that can be connected by time reversal symmetric perturbations $V(q)$, thus enabling single-particle backscattering between different Kramers pairs (Fig. 1$b_2$)[9]. The interference between the incoming and the backscattered edge modes then leads to a pairwise elimination of Kramers pairs[9], and the two-terminal edge conductance drops to $2e^2/h$ in the presence of disorder[1,23–27].

Although many QSHI materials fall into the $n > 1$ Kramers pair regime[17,19,20,22,25] and theoretical studies are extensive[9,23,28,29], experimental reports of backscattering between Kramers pairs remain rare[30], and in QSHI so far even missing. Here, we exploit the interference of incoming and backscattered edge modes and their formation of standing waves that can be readily detected via quasiparticle interference (QPI) in scanning tunneling spectroscopy (STS) (Fig. 1$b_2$)[30]. Investigating the energy-dependent momentum transfer $q$ that characterizes the single-particle backscattering process, we experimentally confirm the pairwise elimination of Kramers pairs in the $\mathcal{S}$-shaped

bandstructure of a QSHI with $n > 1$, effectively restoring the $n = 1$ regime with a single topologically protected Kramers pair[9].

## Results and discussion
### Bulk-boundary correspondence in the QSHI indenene
The QSHI we chose for this study is indenene, a triangular ($1 \times 1$) indium monolayer grown epitaxially on SiC(0001), whose non-trivial topology has been recently demonstrated by independent bulk probes[31,32]. By limiting the surface coverage in a controlled way to about 90% of a monolayer[33], we create holes in the film with clearly defined edge boundaries between indenene and the uncovered SiC surface. These boundaries exhibit three distinct indenene edge types, as illustrated on the left side of Fig. 2a: flat A, flat B and zigzag.

While the zigzag edge bears a resemblance to edges of free-standing honeycomb lattices such as graphene, flat edges A and B terminate the indenene unit cell by either triangle A or B (see inset of Fig. 2a), which are inequivalent due to the carbon atom of the topmost SiC-layer positioned beneath triangle B, but not A[34]. In scanning tunneling microscopy (STM), they are differentiated by gauging the unit cell of indenene vs the position of characteristic subsurface defects, as described in more detail in Supplementary Discussion I. Typically, STM finds straight segments of alternating edge type, as exemplified in Fig. 2a and at the top of Fig. 2b. They are separated by edge imperfections, such as kinks or defects (Fig. 2a right), which serve as scattering centers yielding QPI patterns of edge modes.

STS differential conductance ($dI/dV$) maps essentially reflect the local density of states (LDOS) and, when taken in a small energy interval around the Fermi level, reveal that the indenene edges are clearly metallic (Fig. 2b, bottom). Monitoring the energy dependent LDOS across an indenene edge in Fig. 2d, we find the bulk spectra to reproduce the topological energy gap of $E_{gap} \approx 120$ meV as previously found by angle-resolved photoelectron spectroscopy (ARPES)[31] and GW many-body pertubation theory[34] in Fig. 2c. Approaching the edge, this bulk gap is gradually filled with spectral weight, consistent with the bulk-boundary correspondence and the presence of topological edge modes (Fig. 2b, bottom)[31,32].

We find the integrated in-gap differential conductance to localize exponentially within a penetration depth $\xi = (3.6 \pm 0.6)$ Å from the topographic edge, see Fig. 2e and Supplementary Discussion II. This is in good agreement with the calculated decay constant of $\xi_c = (3.7 \pm 0.3)$ Å when a STS-related orbital sensitivity is taken into account, see Supplementary Discussion III.

### Backscattering of indenene edge states
Having demonstrated the existence of metallic edge modes by STS, let us now turn to their robustness with respect to elastic backscattering. A representative STM topography scan comprising one zigzag and three flat edge segments, interrupted by edge imperfections (white arrows), is shown in Fig. 3a. An STS line scan along the edge, i.e., along the red path outlined in Fig. 3a, is shown in Fig. 3b.

The scan reveals characteristic modulations of the LDOS that are spatially confined to individual edge segments of length $L$. This characteristic particle-in-a-box QPI pattern is a clear indicator of edge modes being reflected back and forth between defects[12,30,35], stabilizing LDOS modulations with wavelengths $2\pi/q$, that submit to the individual resonator condition of each segment. The QPI-active energies significantly exceed the indenene bulk band gap, which is related to the non-linear $\mathcal{S}$-shaped edge state dispersion, as we will see later, and similarly predicted for related materials[22]. This is further supported by strong variations in the level separations $\Delta_j^i = E_i - E_j$ across the wide bandwidth. However, near the bulk gap, we identify almost equidistant level spacings $\Delta_{-2}^{-1}$ and $\Delta_{-3}^{-2}$, evidencing a linear section of the $\mathcal{S}$-shaped edge band dispersion, in which the modes are equally spaced by $\Delta = \hbar v_F^e \pi/L$, with $v_F^e$ being the Fermi velocity of the contributing edge mode.

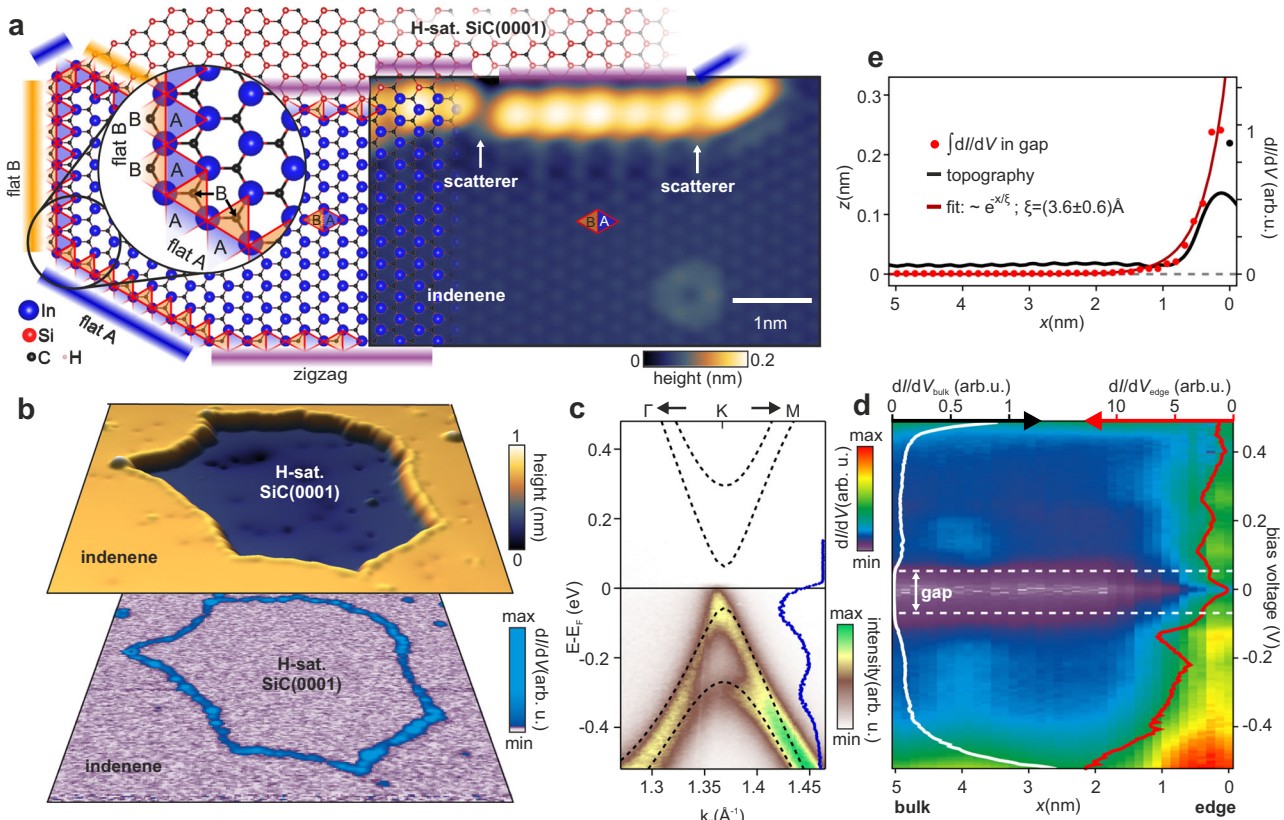

**Fig. 2 | Bulk-boundary correspondence in the QSHI indenene. a** Ball and stick model (left) of indenene on SiC(0001) forming the three edge types (zigzag, flat A, flat B) at the boundary to H-saturated SiC. The inset highlights the carbon position in the top SiC layer, distinguishing flat edge B from flat edge A. STM topography (right) of an indenene edge consisting of zigzag and flat edge A segments, interrupted by kinks that act as scattering centers ($I_T = 300$ pA, $V_{bias} = 0.95$ V). **b** Top: 3D visualization of a $(32$ nm$)^2$ STM topography scan ($I_T = 10$ pA, $V_{bias} = -3$ V) showing a hole with H-saturated SiC in an otherwise closed indenene film. Bottom: Corresponding d$I$/d$V$ map highlighting edge states and taken at constant height by

integrating states within $V_{bias} = (0 \pm 10)$ mV. **c** ARPES measurement and overlaid $G_0W_0$ bandstructure (reprinted from[34]) of indenene's Dirac bands, illustrating band position and charge neutral doping. The inset shows an energy distribution curve at the K-point of indenene. **d** d$I$/d$V$ line scan approaching an indenene zigzag edge, showing that metallic edge states (inset red curve) fill the 120 mV bulk gap (inset white curve). **e** Edge topography (black) and exponential decay of metallic edge states with a decay constant of $\xi = (3.6 \pm 0.6)$ Å fitted to the gap-integrated d$I$/d$V$ signal (red dots). Tunneling parameters of (**d**, **e**) are $I_T = 50$ pA, $V_{bias} = 1$ V, $\delta z = -0.2$ nm (**d**).

Their spatial extension into the bulk is shown in Fig. 3c for selected energy modes $E_{+1}$ to $E_{-3}$ of segment 4, demonstrating that the corresponding LDOS modulations are clearly associated with the edge of the indenene film.

Inspecting the d$I$/d$V$ line spectra of segment 2 in Fig. 3b more closely in Fig. 3d, we further notice the expression of a characteristic energy region $\Delta_{sup}$ between $E_{+1}$ and $E_{-1}$ where d$I$/d$V$ is suppressed. While the size of $\Delta_{sup}$ varies strongly from segment to segment, it always exceeds the size of indenene's band gap $E_{gap} \sim 120$ meV as well as the equidistant energy spacing $\Delta_{-2}^{-1}$ of the adjacent linear band section. Plotting $\Delta_{sup}$ and $\Delta_{-2}^{-1}$ as a function of $1/L$ for a variety of flat A edges of different lengths in Fig. 3f, we find $\Delta_{sup}$ to be universally offset by $\Delta_\infty = (0.26 \pm 0.06)$ eV with respect to the expected particle-in-a-box spacing $\Delta_{-2}^{-1} = \hbar v_F^e \pi / L$, even in the limit of large segment lengths $L \rightarrow \infty$ where $\Delta_{-2}^{-1} \rightarrow 0$ is supposed to vanish. This directly rules out a dynamical Coulomb blockade as origin for the d$I$/d$V$ suppression within $\Delta_{sup}$ (see Supplementary Discussion IV)[36].

Close-up d$I$/d$V$ measurements in Fig. 3e resolve the LDOS within $\Delta_{sup}$ to remain always finite, thus clearly metallic, but essentially featureless except for a zero bias anomaly (ZBA), which suggests second-order effects that will be explored in future studies[37]. Notably, we find no signs related to scattering derived QPI within $\Delta_{sup}$. We thus conclude the presence of metallic edge states within $\Delta_{sup}$ that are resilient to quasiparticle backscattering, as expected for an $n = 1$ QSHI.

Let us now reconcile the edge mode QPI observed in STS with theoretical expectations. For this purpose, we employ density functional theory (DFT) hybrid-functional calculations of bulk indenene[31,34] to construct tight-binding (TB) slabs for all edge types and project the resulting band dispersions onto their respective 1D Brillouin zones. The TB parameters are directly obtained by fitting to bulk DFT calculations, which have previously been demonstrated to accurately reproduce the ARPES spectra and the electronic bulk gap (see Supplementary Discussion III and ref. 31). Results for the indenene flat edge are shown in Fig. 4a, where boundary modes localized at flat edge A are color-coded according to their individual spin character $\langle s_z \rangle$. Modes that localize at the opposite edge, i.e., flat edge B, are shown in dark gray for reference. Detailed calculations of spin and orbital character of all edge terminations are presented in Supplementary Discussion III.

The metallic states at flat edge A arise from two strongly dispersing $\mathcal{S}$-shaped bands with opposite and nearly constant $\langle s_z \rangle$ spin polarization. This overall deviation from the more common linear band scenario (cf. Fig. 1a) is ascribed to the multi-orbital $p_{x,y}$ and $p_z$ character of the edge states, along with the local potential induced by the edge termination, which shifts the TRS-protected crossing point at $\Gamma$ towards higher binding energies, approximately $-0.6$ eV[1,34]. Matching their spin character, the edge states connect the first valence with the second conduction band of the projected 2D band structure (see black bands)[34]. Additionally, a set of Rashba-split free electron

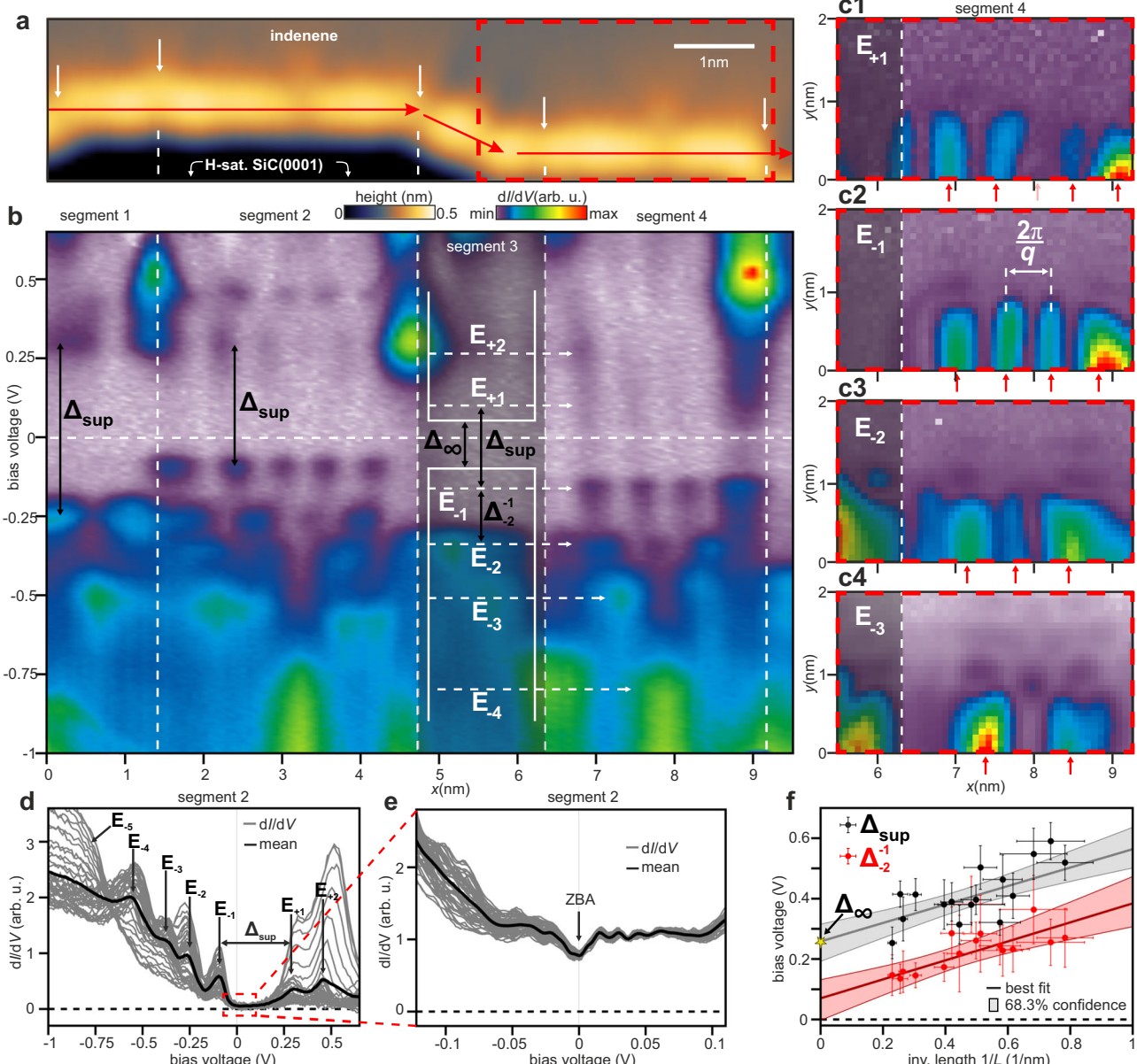

**Fig. 3 | Backscattering in A-terminated indenene edge states. a** Indenene edge topography adjacent to H-sat. SiC, acquired at low corrugation settings ($I_T = 500$ pA, $V_{bias} = -1$ V). Defects and kinks in the edge (white arrows) indicate scattering centers that separate four segments, with segments 1, 2, and 4 being of flat A type, see Supplementary Discussion I and VII. **b** Line scan of d$I$/d$V(E, x)$ taken along the red path in **a** showing typical QPI-related d$I$/d$V$ modulations confined to flat edge A segments as well as variations in $\Delta_{sup}$ among them. **c** Spatially resolved d$I$/d$V(x, y)$ maps of segment 4 (dashed red rectangle in **a**) taken at energies $E_{+1}$ to $E_{-3}$ (labeled in **b**) and demonstrating edge localization of the respective d$I$/d$V$ peaks (red arrows) that are spatially separated by $2\pi/q$ ($I_T = 600$ pA, $V_{bias} = -1$ V). **d** Cumulative plot of segment 2 d$I$/d$V(E)$ curves showing d$I$/d$V$ peaks $E_{-5}$ to $E_{+2}$ as well as an energy region within $\Delta_{sup}$ of QPI suppression. **e** Separate zoom measurement of d$I$/d$V(E)$ in segment 2 demonstrating absence of spatial d$I$/d$V$-modulation except for a mild ZBA as well as finite differential conductance in $\Delta_{sup}$ ($I_T = 250$ pA, $V_{bias} = 0.1$ V). **f** Inverse length $1/L$-dependence of the topmost level spacing $\Delta_{sup}$ and $\Delta_{-2}^{-1}$ in a quantum well picture, as indicated in the inset of **b**. Assuming linear $E(q)$ dependence for these energies, we fit a $\Delta = \Delta_\infty + \hbar v_F^e \pi/L$ dependence to the data yielding $\Delta_\infty = (0.26 \pm 0.06)$ eV for $\Delta_{sup}$.

parabolas centered at the $\overline{X}$ points of the edge Brillouin zone (black arrows) are attributed to the edge and exhibit a dominant $\langle s_y \rangle$ spin polarization (see Supplementary Discussion III).

Clearly, the $\mathcal{S}$-shaped edge dispersion of flat edge A exhibits the qualitative scenario discussed in the context of Fig. 1b, namely an energy interval with only a single ($n = 1$) Kramers pair, and self-overlapping sections of the dispersion with $n > 1$ Kramers pairs, respectively. As described above, the latter allows for intra-band backscattering via characteristic momentum transfers $q$ which connect two different Kramers pairs, as exemplary marked in Fig. 4a. To identify all available TRS-preserving scattering channels $q$ consistent

with the edge dispersion, we apply the T-matrix formalism to calculate the explicit energy-dependent momentum joint density of states (JDOS) $E(q)$ (Fig. 4b), representing an autocorrelation function of the edge bands weighted by scattering matrix elements (see Supplementary Discussion III). Clearly, the favorable scattering channels $q$ disperse along continuous $E(q)$ branches. Within the $n = 1$ energy region where scattering is strictly prohibited by TRS, they exhibit a gap of size $\Delta_\infty \sim 0.25$ eV, in excellent agreement with the $\Delta_\infty = (0.26 \pm 0.06)$ eV that was found in the experimental data of Fig. 3f. Further, extracting the energies $E$ and wavelengths $2\pi/q$ from LDOS modulations in a variety of d$I$/d$V$ line scans (e.g., Fig. 3c top) and plotting the resulting $E(q)$ data

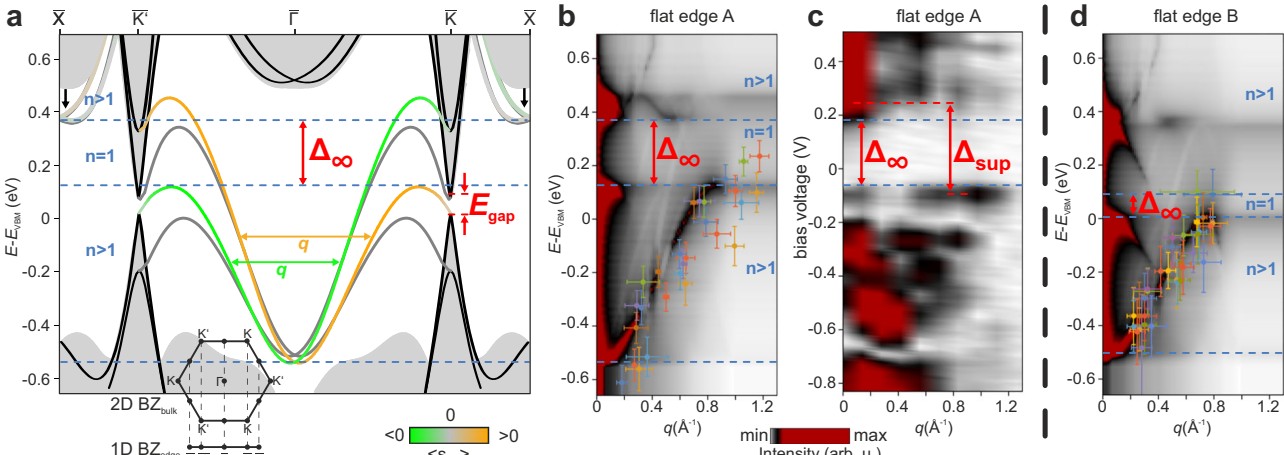

**Fig. 4 | Edge bands and QPI pattern. a** Tight-binding slab calculations (gray) of indenene flat edges with flat edge A localized bands color-coded according to their $\langle s_z \rangle$ character. For clarity, selected bulk bands (black) indicate valence and conduction band onsets with the bulk band gap $E_{gap}$ marked in red. Further indicated for flat edge A are the number of Kramers pairs (blue) in a given energy interval as well as exemplary TRS-allowed backscattering momentum transfers $q$. **b** Energy-dependent momentum JDOS $E(q)$ of $q$ involved in scattering processes that are facilitated by the dispersion of flat edge A, for details see Supplementary Discussion III. Data points are extracted from experimentally observed resonator oscillations (see $2\pi/q$ in Fig. 3c) and offset to compensate for the experimental n-doping, using the valence band maximum (VBM) as a reference point, see "Method" section. Each color represents one edge segment. **c** Modulus of fast Fourier transform of segment 2 in Fig. 3b showing $q$ related spectral weight, QPI suppression (dashed lines) and onset of QPI at positive bias voltages. **d** Calculation of momentum JDOS $E(q)$ associated with flat edge B bands. Experimental data points are extracted from flat edge B segments similar to (**b**). The colorbar in (**b**, **d**, **c**) represents the momentum JDOS and the squared modulus of the FFT, respectively.

points on top of the calculated momentum JDOS $E(q)$ in Fig. 4b, we also find excellent match of theory with experiment. The same applies for the in-gap Fermi velocity $v_F^e$, which we extract both from the slope of the linear region in $E(q)$, as well as in the slope of the data in Fig. 3f, and find $\hbar v_F^e = (0.9 \pm 0.1)$ eV Å and $\hbar v_F^e = (1.0 \pm 0.3)$ eV Å, respectively. Finally, we directly calculate the Fourier transform of the d$I$/d$V$ line scan for segment 2 and plot the result in Fig. 4c for comparison. Despite the spatial confinement of the segment, which produces diffuse spectral features along the $q$-axis of this plot, its overall resemblance to the calculation in Fig. 4b is convincing.

For completeness, we underline this remarkable agreement between experimentally observed and theoretically predicted backscattering further by comparing analogous calculations for flat edge B with experimentally determined $E(q)$ data in Fig. 4d. Again, the TRS protected $n = 1$ region is marked by an—albeit smaller—gap where electron backscattering is prohibited, see Supplementary Discussion V for further details. In contrast to flat edge A, however, the connection of second valence and first conduction bulk band deforms the edge bands near the projected valley momenta $\overline{K}/\overline{K'}$ in such a way, that multiple shorter momentum transfers $q$ are allowed. As the corresponding wavelengths $2\pi/q$ are much longer than the segment lengths available experimentally, their extraction from d$I$/d$V$ lies beyond our capabilities (see Supplementary Discussion V). We note that similar arguments apply to the zigzag edges (see Supplementary Discussion VI).

The spectral properties of indenene edge states laid out in this work reveal elastic single-particle backscattering among different sets of Kramers pairs, driven by the non-linearity of their specific edge dispersions. Consequently, in edge transport measurements of indenene, the presence of disordered defects as scattering centers suppresses the two-terminal conductance to its minimum quantization value of $2e^2/h$[23–27], rather than the purely ballistic value of $n \times 2e^2/h$ expected for $n$ Kramers pairs. This conclusion is of general importance for transport measurements in all QSHIs featuring nonlinear edge band dispersions. Notably, it is especially relevant for WTe$_2$, where band calculations indicate $n = 3$ Kramers pairs[20], while experimental measurements reveal $2e^2/h$[21]. This suggests that two Kramers pairs are eliminated by disorder[23–27], precisely as observed in this work.

## Methods

### Sample preparation
Indenene was grown on N-doped 4H-SiC substrates (12 mm × 2.5 mm) with a resistivity of (0.013) Ωcm (Fig. 2) and (0.015–0.03) Ωcm (Figs. 3, 4). Atomically flat 4H-SiC surfaces were prepared by a dry-etching technique that saturates the silicon dangling bonds with hydrogen and stabilizes the (1 × 1) SiC(0001) surface[38]. Except for data shown in Fig. 2, substrates with a higher dopant concentration are used for practical reasons when performing STM near H-sat. SiC regions, thereby shifting the indenene band gap to negative bias voltages, see Supplementary Discussion II.

Indenene films were grown using a two-step process consisting of hydrogen desorption and indium accumulation in the first step, followed by postannealing in the second step[33]. To increase the indenene edge density on the surface (Fig. 2b), hydrogen-desorption duration (at 600 °C) of 3 min and indium accumulation (at 420 °C) for 25 min is applied in the first step. This preserves ~10% of the H-sat. SiC surface within the otherwise closed indium film. The latter is transformed to indenene in the second growth step by postannealing at 480 °C for 15 min. The substrate temperature was measured with a pyrometer (Keller, detection range 1.1–1.7 µm, emissivity $\epsilon = 85\%$) sensitive to a temperature range of 250–2000 °C. Indium of 99.9999% purity was evaporated from a Knudsen cell held at 770 °C creating an indium flux of ~0.05 Å/s on the substrate surface as measured with a quartz crystal microbalance.

**ARPES measurements** were performed in our home-lab setup equipped with a hemispherical analyzer (PHOIBOS 100), a He-VUV lamp ($\mu$SIRIUS, 21.2 eV), and a 6-axis manipulator capable of LHe-cooling to 20 K. ARPES data shown in Fig. 2c were recorded at 20 K and a base pressure of <10$^{-10}$ mbar. To compensate geometric asymmetry of the experiment, each energy distribution curve is normalized to its integral intensity. Differential pumping of the He-VUV-lamp kept the base pressure below 10$^{-9}$ mbar during the ARPES measurements.

### STM measurements
STM data were acquired at 4.7 K and a base pressure lower than $5 \times 10^{-11}$ mbar (Omicron low-temperature LT STM) using a chemically etched W-tip that was characterized by imaging the Ag(111) surface state. Point spectroscopy d$I$/d$V$ curves were recorded using a standard

lock-in technique with a modulation frequency of 787 Hz and modulation voltage of $V_{rms} = 7$ mV (Fig. 2), $V_{rms} = 1$ mV (Fig. 3e) and $V_{rms} = 10$ mV (Figs. 3b, c, d and 4b, c, d).

**Inverse length analysis** in the quantum well regime (Fig. 3f) is performed by fitting Gaussian profiles to isoenergetic LDOS maxima, allowing us to determine the corresponding bias voltage. The error associated with each peak is given by the Gaussian standard deviation. The segment length $L$ is evaluated based on topography scans, with its error estimated accordingly. Fits to data shown in Fig. 3f are obtained using linear regression.

**$E(q)$-analysis** presented in Fig. 4b, d is determined from the separation between isoenergetic LDOS maxima on the x-axis (indicated in Fig. 3c2), each fitted with Gaussian profiles yielding their x-axis position. The wavelength $2\pi/q$ is then calculated as the average x-separation of neighboring LDOS peaks. Due to the n-doping of indenene films (see Supplementary Discussion II), the energy offset of the valence band onset was determined for each segment and corrected accordingly to allow comparison with TB calculations. The errors for each maximum in the x-direction correspond to their respective Gaussian widths and are propagated to calculate the standard error in $2\pi/q$. Along the energy axis, both the Gaussian width and the offset error are taken into account. For the lowest LDOS maximum, $2\pi/q$ has to be estimated from the quantum-well length and is therefore less precise. For more details see Supplementary Discussion VII.

### DFT and tight-binding calculations of indenene

First-principles calculations are performed using DFT as implemented in the Vienna Ab-initio Simulation Package[39–41]. The exchange-correlation potentials used are the HSE06 hybrid functional[42,43] and PBE[44,45] for bulk and slab calculation respectively, in a non-collinear magnetic moment configuration with self-consistently calculated SOC[46] where specified. The energy cutoff for the plane-wave expansion is set to 500 eV, while the Brillouin zone is sampled on a $12 \times 12 \times 1$ regular mesh.

The bulk structure considered in DFT calculations is the $(1 \times 1)$ reconstruction of a Si-terminated four-layer SiC(0001) substrate with indium atoms in $T_1$ positions. The In-SiC distance is 2.68 Å after converging the forces to within 0.005 eV/Å. A vacuum region in the z-direction of 25 Å is used to disentangle periodic replicas.

TB and $E(q)$ calculations are based on these DFT results and summarized in more detail in Supplementary Discussion III.

### Data availability

The raw data generated in this study have been deposited in the WueData database[47] under accession code https://doi.org/10.58160/9q99u58jb35kmck0.

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

## Acknowledgements

We thank C. Li for helpful discussion. We are grateful for funding support from the Deutsche Forschungsgemeinschaft (DFG, German Research Foundation) under Germany's Excellence Strategy through the Würzburg-Dresden Cluster of Excellence on Complexity and Topology in Quantum Matter ct.qmat (EXC 2147, Project ID 390858490) as well as through the Collaborative Research Center SFB 1170 ToCoTronics (Project ID 258499086). G.P. acknowledges financial support by the European Union—NextGenerationEU, Project code PE0000021—CUP B53C22004060006—"SUPERMOL", "Network 4 Energy Sustainable Transition—NEST" and the European Union—NextGenerationEU under the Italian Ministry of University and Research (MUR) National Innovation Ecosystem grant ECS00000041—VITALITY—CUP E13C22001060006.

## Author contributions

J.E. has realized the epitaxial growth and carried out the photoelectron and scanning tunneling spectroscopy experiments and their analysis. M.I. has conceived the theoretical ideas and performed the DFT and the tight-binding calculations. On the experimental side, contributions came from S.M. and R.C., while F.D., E.M.H., B.T., G.P., D.D.S., and G.S. gave inputs to the theoretical aspects. R.C., G.S., and S.M. supervised this joint project and together with J.E. and M.I. wrote the manuscript with input from all other authors.

## Funding

## Competing interests

The authors declare no competing interests.
