## [Transparent Peer Review file · Nature Communications]

Backscattering in Topological Edge States Despite Time-Reversal Symmetry

Corresponding Author: Professor Ralph Claessen

Version 0:

Reviewer comments:

Reviewer #1

(Remarks to the Author)

The authors present an extensive experimental and theoretical study of backscattering in topological edge states. Of special interest is backscattering between several "S"-shaped edge modes that occur at defects, reducing the system's conductance.

They have grown indenene on a substrate and experimentally explored the edge states with scanning tunneling spectroscopy. To explore backscattering caused by defects, they use a smart design by limiting the surface coverage to about 90% of a monolayer. They find evidence for different defects and measure interference effects from the scattering.

They identify different kinds of edges and see evidence of backscattering surrounding these. They also see scattering at transition points between various edges.

They also compute the electronic states by ab initio calculations and tight-binding calculations.

The first part of the paper is very well written, including the precise Figure 1 explaining the main physical effects.

While topological edge transport is not a new area of research, the present manuscript goes well beyond previous studies in exploring in detail what happens when the dispersion of the edge states is not linear, as in the simplest models. The results are likely to be of interest to researchers in this field.

I, therefore, recommend the publication of the manuscript in Nature Communications.

Some minor comments the authors may consider are:

The joint density of states computed and displayed should be defined in the main article and not just in the supplementary material.

In Figure S1, it is unclear to this reader what sites A and B are in the inset of S1a, since, for example, site A shows a triangle with atomic In corners and seemingly no atoms within the triangle.

Reviewer #2

(Remarks to the Author)

In this manuscript, Jonas Erhardt et al. report the intra-band backscattering within the $n=3$ region of QSHI edge states via the 1D QPI pattern and DFT-tight binding calculation. While the overall interpretation is reasonable and seems to verify a sound but rarely explored physical specific within TRS-protected edge states, there are some issues that need to be clarified prior to the publication in Nature Communications.

Main concerns:

1. Despite the consistency with computational results, the decay length of 3.6 \AA is much shorter than the expected value for

a topological edge state, making people wonder if it stems from topological trivial origin? The authors claim that the non-monotonic dispersion renders the ultrashort decay length, so is the calculated S-shape dispersion consistent with the experimental decay coefficient? Besides, according to the BHZ model, the decay lengths $\propto (dE/dk)/E_{\text{gap}}$ should be energy dependent, owing to the different group velocity characters for energy-selective QPI states. However, this seems to be absent in the experimental data.

2. To fit the calculated JDOS a 0.2 V offset is used to compensate the n-doping, while the relevant arguments based on the bulk gap centered at -0.17 V is confusing (Fig. S2). How the multiple facts of n-doped flat A edge, -0.17 V offset of bulk spectrum, charge-neutral zigzag edge and neural ARPES result reconcile with each other? Especially, under the self-doping effect as approaching the monolayer edge, what's the rationality behind the assignment of 0.2 V doping offset of flat A edge due to the -0.17 V offset of the bulk Fermi level.

3. Does the E_{-1} modulation in Fig. 2 actually come from QPI instead of some bulking or spatial corrugation? There are some modulations in the positive energy with identical periodicity of E_{-1} state, challenging the QPI nature of E_{-1} modulation whose wavelength should be dispersive with respect to energy.

4. The excellent agreement between experimental and computational value of Δ_{∞} constitutes important evidence. Therefore, the choice of λ_{SOC} parameter and corresponding reasonability should be elaborated since its value directly determines the splitting of K valley and hence, the size of Δ_{∞} .

Minor comments:

1. The title of main text is inconsistent with the one in supplementary information.

2. I prefer the title in SI since there weren't any phenomenon that violates the fundamental TRS principle.

3. In abstract and main text, the statements like "pairwise coupling between energy-degenerate Kramer pairs" are misleading as they seem to express that the backscattering couples the TRS-guaranteed Kramer degeneracies.

4. It's better to add the color bar for dI/dV maps/line scans.

Reviewer #3

(Remarks to the Author)

This manuscript presents a compelling experimental and theoretical study on backscattering in topological edge states of the quantum spin Hall insulator (QSHI) indenene. The authors present a significant advance in the understanding of topological edge states, demonstrating that backscattering can occur even in the presence of time-reversal symmetry when multiple Kramers pairs are present. The findings are significant for the field of topological materials and have broad implications for spintronic applications. The manuscript is suitable for publication in Nature Communications after addressing the following comments.

1. The term quantum spin Hall insulator (QSHI) is conventionally reserved for systems with a topological invariant that ensures robust, backscattering-immune helical edge states protected by time-reversal symmetry. However, the observed $n > 1$ regime, where backscattering between Kramers pairs becomes possible, challenges this strict definition. The authors should explicitly address whether such systems still qualify as QSHIs or if an alternative classification would be more appropriate.

2. While the evidence for elastic single-particle backscattering is compelling, the authors should briefly consider whether other mechanisms could contribute to the observed QPI patterns. For instance, Phys. Rev. B 103, 235164 (2021) and Phys. Rev. Lett. 121, 106601 (2018).

3. The manuscript primarily focuses on flat edge A, while flat edge B and zigzag edges receive limited attention. A more detailed comparison of backscattering behavior across all edge terminations, including their respective momentum-transfer dependencies and energy gaps—would provide a more comprehensive understanding of edge-state robustness. For example, does the reduced gap in flat edge B correlate with stronger substrate coupling or enhanced disorder susceptibility?

4. The substrate could introduce additional scattering channels. Are the observed backscattering features intrinsic to indenene, or could hybridized bands with the substrate play a role? The authors should comment on whether substrate effects were accounted for in their theoretical models and how such interactions might influence the edge-state dispersion and protection.

Version 1:

Reviewer comments:

Reviewer #1

(Remarks to the Author)

The authors have given a satisfactory response to my first report.

I recommend publication in Nature Communications.

Reviewer #2

(Remarks to the Author)

All my concerns have been addressed and I recommend the publication in Nature Communications.

Reviewer #3

(Remarks to the Author)

The authors have addressed all my comments. Now I recommend the publication of this paper in Nature Communications.

Backscattering in Topological Edge States Despite Time-Reversal Symmetry - Response to Reviewers -

June 25, 2025

Reviewers' comments to the Authors:

Reviewer #1 (Remarks to the Author):

The authors present an extensive experimental and theoretical study of backscattering in topological edge states. Of special interest is backscattering between several "S"-shaped edge modes that occur at defects, reducing the system's conductance. They have grown indenene on a substrate and experimentally explored the edge states with scanning tunneling spectroscopy. To explore backscattering caused by defects, they use a smart design by limiting the surface coverage to about 90% of a monolayer. They find evidence for different defects and measure interference effects from the scattering. They identify different kinds of edges and see evidence of backscattering surrounding these. They also see scattering at transition points between various edges. They also compute the electronic states by ab initio calculations and tight-binding calculations. The first part of the paper is very well written, including the precise Figure 1 explaining the main physical effects. While topological edge transport is not a new area of research, the present manuscript goes well beyond previous studies in exploring in detail what happens when the dispersion of the edge states is not linear, as in the simplest models. The results are likely to be of interest to researchers in this field.

I, therefore, recommend the publication of the manuscript in Nature Communications.

Some minor comments the authors may consider are:

The joint density of states computed and displayed should be defined in the main article and not just in the supplementary material. In Figure S1, it is unclear to this reader what sites A and B are in the inset of S1a, since, for example, site A shows a triangle with atomic In corners and seemingly no atoms within the triangle.

Response:

We thank the reviewer for his/her appreciation of our work and for recommending it for publication.

Regarding the reviewer's suggestion, we prefer to mention the definition of the momentum joint density of states in the Supplementary Information, as this relation was established in previous studies and not developed in our work.

We also thank the reviewer for pointing out the missing explanation of A and B in Fig. S1. We have now added this clarification to the corresponding section. For improved clarity, we replaced the term "site A and B" with "positions A and B," as these do not correspond to atomic sites of the indenene bulk unit cell but rather to positions within the unit cell that are subject to different substrate potentials. Specifically, position B is located above the carbon atom of the topmost SiC layer, which induces a local potential in that half of the indenene unit cell. In contrast, position A lies above a void of this SiC layer and is therefore not subject to such a potential.

Reviewer #2 (Remarks to the Author):

In this manuscript, Jonas Erhardt et al. report the intra-band backscattering within the $n=3$ region of QSHI edge states via the 1D QPI pattern and DFT-tight binding calculation. While the overall interpretation is reasonable and seems to verify a sound but rarely explored physical specific within TRS-protected edge states, there are some issues that need to be clarified prior to the publication in Nature Communications.

Response:

We thank the reviewer for their appreciation of our work and for the insightful comments, which have helped us improve the manuscript in this revised version.

Main concerns:

Comment 1:

Despite the consistency with computational results, the decay length of 3.6\AA is much shorter than the expected value for a topological edge state, making people wonder if it stems from topological trivial origin? The authors claim that the non-monotonic dispersion renders the ultrashort decay length, so is the calculated S-shape dispersion consistent with the experimental decay coefficient? Besides, according to the BHZ model, the decay lengths $\propto (dE/dk)/E_{\text{gap}}$ should be energy dependent, owing to the different group velocity characters for energy-selective QPI states. However, this seems to be absent in the experimental data.

Response:

We thank the reviewer for highlighting the need to clarify our discussion of the edge state decay length, which we have addressed in the revised manuscript. Our key point in this discussion was to emphasize that the decay length of topological edge states – particularly in the case of non-linear dispersions – does not necessarily follow the simple relation derived in the BHZ model for linear edge states localized near the Γ point of their 1D Brillouin zone. Regarding the quantitative scale of the decay constant, we would like to stress that the decay constants observed for indenene is fully consistent with topological edge states, as similarly reported for related materials [1, 2, 3]. These studies suggest a connection between the decay length and a non-linear edge band dispersion spanning the entire 1D Brillouin zone. However, we understand that our initial discussion following this suggestion may have been misleading, and we have therefore removed the comparison to the BHZ relation.

The experimentally observed in-gap decay coefficients are consistent with the calculated values for the S-shaped dispersion when the orbital extension into the vacuum is taken into account. This consideration is necessary for the comparison due to the presence of multiple Kramers pairs with different orbital character (see Fig. S6), as well as the inherent sensitivity of STS to electronic states with out-of-plane character [4]. These orbitals extend further into the vacuum than in-plane states, leading to a larger overlap with the tip wavefunction and thus dominate the charge profile detected in STS. We have added a corresponding description of the calculated decay constant to Supplementary Discussion V and Fig. S5b,c.

Regarding the absence of an energy dependence of the decay constant in the QPI states (Fig. 3c), we first note that these states mainly originate from the linear section of the \mathcal{S} -shaped band and thus share similar group velocities, as inferred from the equal level spacing and the extracted dispersion (Fig. 4). Second, a simple relationship to the group velocity – such as the one derived in the BHZ model – cannot be established for non-BHZ-like dispersions and likely involves more complex parameterization [3]. Given the unknown scaling behavior for the decay constant, and considering the momentum-integrating and orbital-selective nature of STS in the presence of multiple Kramers pairs, we refrain from speculating on a group-velocity-induced energy dependence in the QPI state decay constant.

Comment 2:

To fit the calculated JDOS a 0.2 V offset is used to compensate the n-doping, while the relevant arguments based on the bulk gap centered at -0.17 V is confusing (Fig. S2). How the multiple facts of n-doped flat A edge, -0.17 V offset of bulk spectrum, charge-neutral zigzag edge and neural ARPES result reconcile with each other? Especially, under the self-doping effect as approaching the monolayer

edge, what's the rationality behind the assignment of 0.2 V doping offset of flat A edge due to the -0.17 V offset of the bulk Fermi level.

Response:

We thank the reviewer for his/her comment and have added a corresponding statement about the energy offset in the revised manuscript. To clarify, we employed 4H-SiC(0001) substrates with two different n-doping levels, each optimized for specific parts of the study as noted in the original Method section. For demonstrating metallic edge states in Fig. 2, we used a doping level yielding charge-neutral indenene, as confirmed by ARPES and STS measurements in Fig. 2. However, this SiC doping caused tip instabilities when imaging H-saturated SiC regions at atomic resolution closer to the Fermi level, complicating detailed STS studies along the indenene edge such as those shown in Fig. 3 and Fig. 4. For practical reasons, we thus switched to a higher SiC doping level that enabled atomic-resolution imaging of H-sat. SiC (see Fig. S3b₄), though this shifted the indenene gap center to approximately -0.17 eV (see Fig. S2).

To compensate for this energy shift when comparing with the calculations in the old version of the manuscript, a global correction of 0.2 eV was applied to the QPI levels investigated on samples with identical nominal substrate doping level. The discrepancy to -0.17 eV noted by the reviewer stems from the off-centered Fermi level in the calculations, see old version of Fig. 4a.

Considering the comment of the reviewer, instead of applying a global offset, a more robust analysis involves determining the offset for each edge segment individually, which we have implemented in the revised version and highlighted in the Method section. To this end, we selected the valence band maximum as a clearer reference point in both theoretical calculations and dI/dV spectra, and added an exemplary description to the Supplementary Discussion II. Regarding self-doping effects upon approaching the edge, it is difficult to reliably trace bulk band positions in STS, which is why we integrated such effects in the estimated error of the offset values.

Comment 3:

Does the E_{-1} modulation in Fig. 3 actually come from QPI instead of some bulking or spatial corrugation? There are some modulations in the positive energy with identical periodicity of E_{-1} state, challenging the QPI nature of E_{-1} modulation whose wavelength should be dispersive with respect to energy.

Response:

As correctly pointed out by the reviewer the periodicity of E_{-1} and E_{+1} in segment 4 of Fig. 3b appears to be quite similar. To exclude topographic imprints when recording the dI/dV spectra, we stabilized the tip – by disabling the feedback loop – at a bias voltage where the topography appears approximately featureless, such as the case at -1 V at which Fig. 3a and b are stabilized.

Rather than arising from topographic effects, the modulations mentioned by the reviewer can be attributed to scattering and subsequent interference processes, as revealed by the JDOS of flat edge A in Fig. 4b. In the positive bias voltage region, available scattering vectors q range from 0.3\AA^{-1} to 0.7\AA^{-1} , supporting standing wave patterns with periodicities of approximately $2\pi/q \sim 0.9\text{ nm}$ – consistent with the periodicity seen at positive bias voltages in segment 4 of Fig. 3b. The reduced modulation intensity observed in Fig. 3b correlates with the lower momentum JDOS values for these q -vectors. Additionally, in other segments such as segment 2, the energy levels E_{-1} and E_{+1} are less aligned compared to segment 4, due to differing resonator conditions. We therefore conclude that both modulations originate from the momentum-resolved JDOS of the flat edge A band structure.

Comment 4:

The excellent agreement between experimental and computational value of Δ_∞ constitutes important evidence. Therefore, the choice of λ_{SOC} parameter and corresponding reasonability should be elaborated since its value directly determines the splitting of K valley and hence, the size of Δ_∞ .

Response:

In response to the reviewer's comment, we have elaborated on the choice of λ_{SOC} in the main text and added a table summarizing all tight-binding parameters, obtained from fits to the DFT band

structure, to Supplementary Section V. The fitted value of λ_{SOC} is in excellent agreement with the K-point band splitting between the second valence band and the first conduction band observed in ARPES [5], which directly reflects λ_{SOC} in indenene [5, 6].

Minor comments:

1. The title of main text is inconsistent with the one in supplementary information.

Response: We thank for indicating this inconsistency and corrected it.

2. I prefer the title in SI since there weren't any phenomenon that violates the fundamental TRS principle.

Response: We appreciate the reviewer's opinion, but prefer to retain the current title, exactly *because* there are no TRS violating phenomena involved in the reported backscattering processes.

3. In abstract and main text, the statements like "pairwise coupling between energy-degenerate Kramer pairs" are misleading as they seem to express that the backscattering couples the TRS-guaranteed Kramer degeneracies.

Response: We thank the reviewer for pointing out these misleading formulations and have revised the corresponding text passages accordingly.

4. It's better to add the color bar for dI/dV maps/line scans.

Response: We followed the reviewer's suggestion and added color bars to the respective panels.

Reviewer #3 (Remarks to the Author):

This manuscript presents a compelling experimental and theoretical study on backscattering in topological edge states of the quantum spin Hall insulator (QSHI) indenene. The authors present a significant advance in the understanding of topological edge states, demonstrating that backscattering can occur even in the presence of time-reversal symmetry when multiple Kramers pairs are present. The findings are significant for the field of topological materials and have broad implications for spintronic applications. The manuscript is suitable for publication in Nature Communications after addressing the following comments.

Response:

We are grateful for the referee's positive assessment of our work and thank him/her for the insightful comments that have supported the refinement of the revised version of the manuscript.

Comment 1:

The term quantum spin Hall insulator (QSHI) is conventionally reserved for systems with a topological invariant that ensures robust, backscattering-immune helical edge states protected by time-reversal symmetry. However, the observed $n > 1$ regime, where backscattering between Kramers pairs becomes possible, challenges this strict definition. The authors should explicitly address whether such systems still qualify as QSHIs or if an alternative classification would be more appropriate.

Response:

As the reviewer correctly noted, the QSHI phase is distinguished from topologically trivial insulators by a \mathcal{Z}_2 topological invariant. However, this classification is entirely determined by the bulk electronic structure and neither requires the presence of edges nor involves the edge state properties mentioned by the reviewer [7]. When a QSHI is truncated, time-reversal symmetry (TRS) indeed protects the existence of at least one of an odd number n of metallic Kramers pairs in the bulk gap by forbidding elastic single-particle backscattering. Yet, this affects only backscattering between edge modes within the same Kramers pair, and leaves coupling of different Kramers pairs for $n > 1$ allowed, as observed in this study.

Nevertheless, also in this $n > 1$ case TRS preserves the intriguing properties of the QSHI edge states by employing the fact that the number n of Kramers pairs is odd within the bulk band gap [8, 7, 9]: Since elastic single-particle backscattering can only localize an even number of Kramers pairs [9, 10], one helical pair always remains protected by TRS against backscattering and is thus perfectly transmitted at non-magnetic defects – just as in the $n = 1$ case, making an alternative classification for the $n > 1$ case unnecessary. We have followed the reviewer's suggestion and emphasized the importance of the \mathcal{Z}_2 invariant in the revised manuscript.

Comment 2:

While the evidence for elastic single-particle backscattering is compelling, the authors should briefly consider whether other mechanisms could contribute to the observed QPI patterns. For instance, Phys. Rev. B 103, 235164 (2021) and Phys. Rev. Lett. 121, 106601 (2018).

Response:

As correctly pointed out by the reviewer, other scattering mechanisms are in principle allowed to affect the topological edge states and its conductance quantization [11, 12, 13, 14, 15]. However, we confidently attribute the observed QPI standing waves to elastic single-particle backscattering, for the following reasons.

The formation of standing waves in QPI fundamentally relies on the coherence between incoming and backscattered states [16, 17], making this technique effectively insensitive to incoherent and inelastic scattering processes, such as discussed, for example, in Ref. [14]. Regarding the second mechanism involving quasielastic backscattering mentioned by the reviewer, it remains unclear to what extent coherence is preserved in this quasielastic process [15]. Even if coherence were preserved, we note that the observed QPI signal is suppressed within the $n = 1$ Kramers pair interval Δ_∞ , a regime where the alternative mechanisms would still be expected to contribute, while only elastic single-particle backscattering is suppressed. We thus conclude that the observed QPI is related to elastic single-particle backscattering.

Comment 3:

The manuscript primarily focuses on flat edge A, while flat edge B and zigzag edges receive limited attention. A more detailed comparison of backscattering behavior across all edge terminations, including their respective momentum-transfer dependencies and energy gaps—would provide a more comprehensive understanding of edge-state robustness. For example, does the reduced gap in flat edge B correlate with stronger substrate coupling or enhanced disorder susceptibility?

Response:

We thank the reviewer for the thoughtful suggestion to compare the different edge types of indenene in more detail. In response, we have added a discussion of the flat edge B along with details on the substrate-relation of its reduced $n = 1$ Kramers pair interval Δ_∞ to the Supplemental Information (Supplemental Section VI), complementing the existing discussion of the zigzag edge in Supplemental Section VII. The corresponding momentum JDOS calculations reveal that experimental length limitations prevent the formation of well-defined quantum well states in the available zigzag segments (Fig. S8e), and also hinder precise determination of the $n = 1$ Kramers pair interval Δ_∞ for flat edge B (Fig. S7c). With respect to disorder susceptibility, no general conclusion can be drawn due to the inherently unknown nature of the scattering potential provided by the defects. Nevertheless, analysis of the occupied quantum well states on flat edge B remains feasible and shows good agreement with our tight-binding calculations (Fig. 4d and Supplemental Section VI).

We, however, want to emphasize that our primary goal is to experimentally demonstrate elastic single-particle backscattering in QSHI edge states from a general perspective. Due to the mentioned length L constraints of edge segments accessible in experiments, we focused on flat edge A in the main text, which provides many q -vectors exceeding the minimal $q_{\min} = 2\pi/L$, ideal for identifying quantum well states and the corresponding energy suppression region Δ_∞ .

Comment 4:

The substrate could introduce additional scattering channels. Are the observed backscattering features intrinsic to indenene, or could hybridized bands with the substrate play a role? The authors should comment on whether substrate effects were accounted for in their theoretical models and how such interactions might influence the edge-state dispersion and protection.

Response:

We thank the reviewer for pointing out a missing statement regarding potential substrate hybridization. Such effects can be safely excluded for the indenene edge states, as their bands lie entirely within the wide band gap of the 4H-SiC(0001) substrate, as shown in the band structure calculations in Supplementary Figure 3 of Ref. [5]. To reinforce this point in our study, we have added a corresponding statement along with a representative dI/dV spectrum acquired on the H-saturated 4H-SiC(0001) surface next to the indenene film to Supplementary Section III (Fig. S3a). The measured dI/dV spectrum confirms the 4H-SiC gap as well as the absence of SiC states in the energy range from -1 V to 1 V, thereby ruling out hybridization with the indenene edge states in this energy window.

References

- [1] Ok, S. *et al.* Custodial glide symmetry of quantum spin Hall edge modes in monolayer WTe₂. *Phys. Rev. B* **99**, 121105 (2019).
- [2] Wada, M., Murakami, S., Freimuth, F. & Bihlmayer, G. Localized edge states in two-dimensional topological insulators: Ultrathin Bi films. *Phys. Rev. B* **83**, 121310 (2011).
- [3] Bieniek, M., Väyrynen, J. I., Li, G., Neupert, T. & Thomale, R. Theory of glide symmetry protected helical edge states in a WTe₂ monolayer. *Phys. Rev. B* **107**, 195105 (2023).
- [4] Wiesendanger, R. *Scanning Probe Microscopy and Spectroscopy Methods and Applications* (Cambridge University Press, UK-CB2 2RU Cambridge, 1998).
- [5] Bauernfeind, M. *et al.* Design and realization of topological dirac fermions on a triangular lattice. *Nat. Commun.* **12**, 5396 (2021).
- [6] Eck, P. *et al.* Real-space obstruction in quantum spin Hall insulators. *Phys. Rev. B* **106**, 195143 (2022).
- [7] Kane, C. L. & Mele, E. J. Z₂ Topological Order and the Quantum Spin Hall Effect. *Phys. Rev. Lett.* **95**, 146802 (2005).
- [8] Kane, C. L. & Mele, E. J. Quantum Spin Hall Effect in Graphene. *Phys. Rev. Lett.* **95**, 226801 (2005).
- [9] Qi, X.-L. & Zhang, S.-C. Topological insulators and superconductors. *Rev. Mod. Phys.* **83**, 1057–1110 (2011).
- [10] Wu, C., Bernevig, B. A. & Zhang, S.-C. Helical Liquid and the Edge of Quantum Spin Hall Systems. *Phys. Rev. Lett.* **96**, 106401 (2006).
- [11] Crépin, F. m. c., Budich, J. C., Dolcini, F., Recher, P. & Trauzettel, B. Renormalization group approach for the scattering off a single Rashba impurity in a helical liquid. *Phys. Rev. B* **86**, 121106 (2012).
- [12] Schmidt, T. L., Rachel, S., von Oppen, F. & Glazman, L. I. Inelastic electron backscattering in a generic helical edge channel. *Phys. Rev. Lett.* **108**, 156402 (2012).
- [13] Budich, J. C., Dolcini, F., Recher, P. & Trauzettel, B. Phonon-induced backscattering in helical edge states. *Phys. Rev. Lett.* **108**, 086602 (2012).
- [14] Väyrynen, J. I., Pikulin, D. I. & Alicea, J. Noise-Induced Backscattering in a Quantum Spin Hall Edge. *Phys. Rev. Lett.* **121**, 106601 (2018).
- [15] McGinley, M. & Cooper, N. R. Elastic backscattering of quantum spin Hall edge modes from Coulomb interactions with nonmagnetic impurities. *Phys. Rev. B* **103**, 235164 (2021).
- [16] Crommie, M. F., Lutz, C. P. & Eigler, D. M. Imaging standing waves in a two-dimensional electron gas. *Nature* **363**, 524–527 (1993).
- [17] Hasegawa, Y. & Avouris, P. Direct observation of standing wave formation at surface steps using scanning tunneling spectroscopy. *Phys. Rev. Lett.* **71**, 1071–1074 (1993).